# Evolution and Transformation Analysis of Land-use in Mountainous "Granary"—Evidence from Typical Basin in Karst Mountainous Areas of Southwest China

**Mei Chen, Yangbing Li \*, Yiyi Zhang, Limin Yu and Linyu Yang**

School of Geography and Environmental Sciences, Guizhou Normal University, Huaxi District, Guiyang 550025, China; mui-chen@gznu.edu.cn (M.C.); 20030090031@gznu.edu.cn (Y.Z.); 21010090324@gznu.edu.cn (L.Y.); 222100090325@gznu.edu.cn (L.Y.)

**\*** Correspondence: li-yabin@gznu.edu.cn; Tel.: +86-1388-5141-624

**Abstract:** Intermountain basins are the granaries of the karst mountains of southwest China. Revealing the process and trend of land-use transformation in typical mountainous "granaries" is of great significance to optimizing land-use, food security, and rural revitalization in the karst mountains of southwest China. Lianjiang basin in Huishui County is the largest in Guizhou Province of southwest China, and a typical mountainous "granary" is used as a case study area. Based on seven periods of high-definition remote sensing images from 1966 to 2020 and on-the-spot investigation, we adopt the analysis methods of landscape pattern, moving window, and buffer zone and conduct an in-depth study on the spatial evolution of land-use and the land-use transformation law of the typical "granaries" of karst mountainous areas in southwest China from the perspectives of changes in the quantity of land-use, changes in the types of land-use, and changes in the landscape pattern. The results showed that the transformation of land-use in the basin was mainly manifested in the transformation of the function of cultivated land and the diversification of the function of cultivated land. The landscape pattern of the basin was gradually transformed from traditional rural landscape to modern agricultural landscape. The gradient effect of land-use evolution and transformation in the study basin was obvious. Traffic, location, and land resources significantly affect land-use transformation in the basin. Based on the above analysis, this paper proposes that the "granary" basins in karst mountainous areas should optimize the land-use of the basin by the trade-off between regional socio-economic development, ecological protection, and food security. The results of this paper provide a reference for land-use optimization and rural development in the granary-type basins and other similar areas in the karst mountains of southwest China.

**Keywords:** land-use; transition; cultivated land; basin; Guizhou province

## 1. Introduction

Land-use activities have changed much of the land surface [1], with about 33% of the land now used for agriculture [2]. Different land cover trajectories exist in different regions of the world [3]. Asia has concentrated areas with the fastest land cover change, especially dry land degradation. For example, cultivated land in the southeastern United States and eastern China is rapidly decreasing [4]. Therefore, land change science has become a fundamental component of global environmental change and sustainable research [5]. Land-use transition refers to the process in which a region changes from one form of land-use to another within a period, driven by social and economic changes and innovations [6,7]. The land-use form includes spatial and functional structures, and the land-use transition effect includes social, economic, and environmental aspects [8,9]. Different economic and social development stages correspond to different regional land-use patterns and land-use transition stages [1,10,11]. Based on the unique perspective of land-use transition, it is of great significance to prospectively study the impact of the urbanization process on

the environment of agricultural areas and its control methods [12,13]. The transition of land-use function is one of the important manifestations of land-use transition and one of the critical entry points for the study of land-use transition [14,15]. Relevant scholars have rich research results on this, and found that the regional spatial pattern of land-use change in China was basically the same in 2010–2015 compared with 2000–2010, with the expansion of construction land in the west accelerating significantly, the growth rate of cultivated land area accelerating further, and the rate of reduction of forest and grassland area increasing [16,17]. Studies have shown that the function of cultivated land in China generally began to transform in 2006 [18], and the function of cultivated land-use in the Pearl River Delta has realized the transition from economic and social to ecological and economic [19]. The transition of rural land-use in mountainous areas reflects the trend turning point in the long-term change of land-use patterns [20,21]. Unlike the plains, the topography is the most important factor affecting the evolution of cultivated land in mountainous areas [22]. However, current research is often based on county and other macro scales and mainly focuses on eastern China [23–25]. There is insufficient research on land-use transformation and its further effects on special landform types in the mountainous regions of Southwest China.

The karst mountains in southwest China are dominated by the Yungui Plateau, and the northern, eastern, and southeastern parts of the country are the slopes transitioning to the Sichuan Basin, hilly areas of Pan-Western-Hunan and the Guangxi Basin, respectively, with medium and low mountains dominating. Guizhou Province is the center of the karst mountainous areas in southwest China, and the land resources are more restricted [26]. Therefore, there is a certain area of flat land in the mountains with good soil quality and irrigation conditions; the local custom is called 'basin' [27]; it is an important "granary" base in Guizhou Province, and the province's stable grain-producing areas are mainly distributed in a small number of relatively flat "basins" in the mountains. On the one hand, under the background of more mountains and less land, rural basins in karst mountains are facing multiple choices of land-use and multi-path evolution due to their land-use suitability. On the other hand, under the background of Western development, with the impact of urbanization, industrialization, and urban–rural integration on rural areas, and under the influence of multiple backgrounds, such as the promotion of new rural construction and rural economic transition and development, in recent years the landscape and land-use pattern of rural basins in karst mountains have undergone dramatic changes [28]. Some contiguous basin lands with good agricultural production conditions are constantly occupied by construction land. In particular, the cultivated land landscape of some "10,000 mu (mu, the Chinese unit of land measurement, commonly 666.7 square meters) cultivated land basin" is gradually disappearing. At the same time, some basins have developed modern agriculture with special characteristics and leisure agriculture through the transition of agricultural land-use (land circulation, planting structure adjustment). This land-use is shifting from traditional agriculture to modern agriculture. It has shifted to the cultivation of cash crops, even tourism, and ecological utilization. Therefore, under the natural background of karst mountains with many hills and less flat land, and the current socio-economic background of rapid urbanization, it is very necessary to understand in depth what changes in land-use in the rural basins of karst mountains have occurred, and why these changes have occurred. What is the changing spatial and temporal pattern? What are the spatial differences in the transition of cultivated land function? How to respond to such changes to optimize control? Answering these questions is very urgent at present. Existing studies on the basin in karst mountainous areas focus on mountain-basin land-use coupling [29], the evolution of basin land-use functions [30], transformation of paddy land-use in intermountain basins [31] and differences in urban patterns in intermountain basins [27,32]. However, there is still a lack of in-depth research, based on typical cases, on the changing law of land-use transition in rural basins over a longer period of time and at multiple spatial scales.

In summary, revealing the process and trend of land-use change in the "granary" basin in karst mountainous areas and its driving mechanism will help provide references for optimizing land-use, sustainable development, rural revitalization, and modern agricultural development in the basin, and provide microscale cases for land-use transition theory. This paper selects the Lianjiang Basin in Huishui County, the largest Guizhou Province basin, an important "granary" base in typical karst mountainous areas, as the study area. The aim is to conduct an in-depth study of the spatial and temporal evolution of land-use and land-use transformation patterns from the perspectives of quantitative land-use changes, the perspective of type change, and the perspective of a landscape pattern change, and at multiple spatial scales, based on a long-term series of high-definition remote sensing images and field investigations. The study results provide references for optimizing land-use, food security, and coordinated development of human–land relationships in rural basins in karst mountainous areas and other similar areas.

## 2. Study Area

The research area, the Lianjiang Basin, is located in the southern Guizhou Buyi and Miao Autonomous Prefecture in the south of Guizhou Province, the north is close to the Huaxi District of Guiyang, the capital city of Guizhou Province, the west is close to Changshun county, and the basin mainly includes Gaozhen, Heping town, Sandu town, and Haohuahong town, with a total area of 85.71 km$^2$; it is the economic development center of Huishui county (Figure 1). In the interior of this basin, the Lianjiang River runs through the entire basin area from north to south, so it is named after the Lianjiang Basin. The basin's topography is high in the north and low in the south. It is the location of the Lianjiang alluvial plain. The land is fertile and has a subtropical monsoon climate. The average annual temperature is 15.8 °C, and the average annual rainfall is 1213.4 mm. The basin has long been planted mainly with traditional crops, such as rice, and is one of the important "granary" bases in Guizhou Province. However, in recent years, due to the low economic income from traditional food crops, most of the traditional agricultural cultivation in the basin has gradually shifted to modern agriculture, such as greenhouse vegetables, fruit bases, and the flower and seedling industry, resulting in significant changes of land-use in the basin [31].

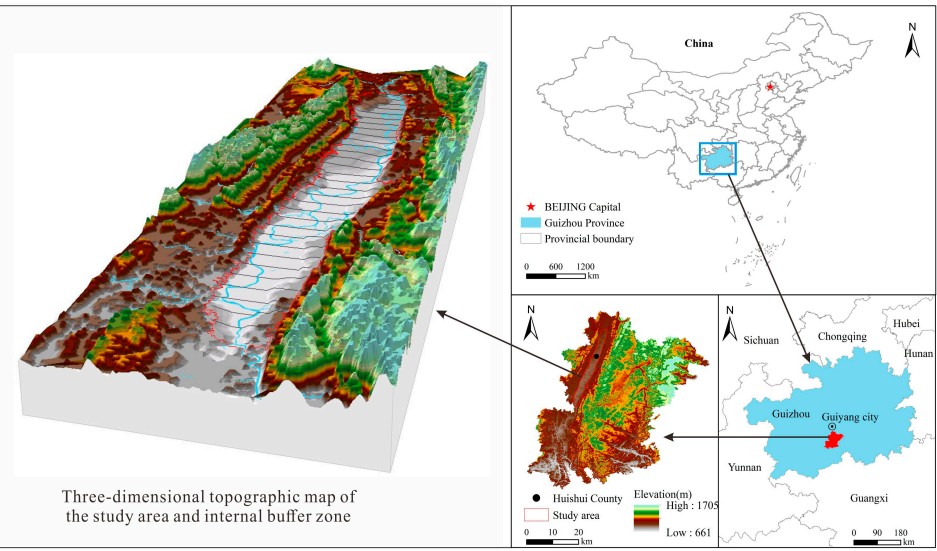

**Figure 1.** Topography and location of the study area.

## 3. Research Methods

### 3.1. Research Ideas

The diversity of social needs, the multi-suitability of land-use, and the plurality of regional development goals have led to diversifying the function of arable land-use [33].

Therefore, how to rationally allocate arable land resources according to the many functions of arable land, to meet the diversified needs of human society for the products and services formed on arable land is the difficulty, and is a hotspot in current arable land protection work [34]. Based on existing studies [31], the authors argue that, during the evolution of the land-use in the basin, the general trend is that the arable land-use function gradually decreases while other functions gradually increase, or although the arable land-use function is still the main feature, the diversity of arable land function continues to increase. Suppose the evolution of the basin's land-use function is inconsistent with this general trend. In that case, it may be due to natural and socio-economic factors affecting the basin's land-use function change. Therefore, the land-use transition of the basin is mainly reflected in two aspects: (1) Cultivated land function transfer: Cultivated land is lost due to conversion to construction land, such as residential and industrial and mining storage, transportation, and other land, the amount of cultivated land is reduced, agricultural space is continuously occupied, and cultivated land is converted into non-agricultural land; (2) Increased versatility of arable land: The functions of arable land-use have gradually diversified, shifting from the cultivation of grain crops to the cultivation of non-grain crops, and the formation of large-scale agricultural land [35]. Traditional agricultural space has gradually evolved into modern agricultural space, such as sightseeing agriculture. The above evolution and transformation of land-use in rural basins can be reflected through the spatial and temporal changes in the quantitative aspect of land-use, land-use types, and spatial patterns of land-use landscapes in rural basins, and the formation of different land-use transformation processes and transformation trends in different natural and socio-economic buffers (Figure 2).

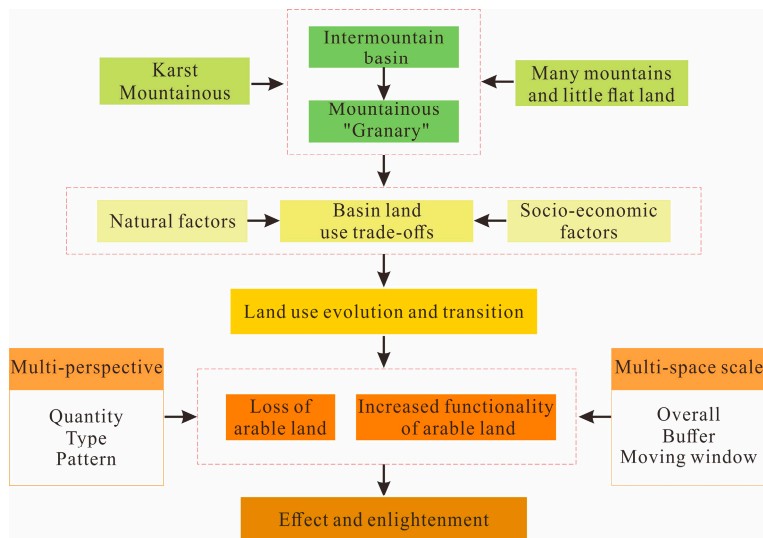

**Figure 2.** A research framework for the mountain granary's evolution and land-use transition.

### 3.2. Data Sources and Land-use Classifications

The sources of land-use data in different periods for the Lianjiang Basin are shown in Table 1. The research periods were determined mainly by considering the accessibility of high-resolution data in the research area and when land-use in the study area changed significantly. ENVI 5.3 and ArcGIS10.6 software were utilized for projection transformation and geometric correction of multi-source images of the study area. Human–computer interaction interpretation was used to obtain each period's land-use vector data. Regarding the ground data of the same period, some sample areas were randomly selected, and through several field investigations, the classification confusion matrix was statistically obtained, and the Kappa index was calculated to test the accuracy of the land-use data in each period. The results showed that the Kappa indices for 1966, 1973, 1990, 2006, 2010, 2016, and 2020 were 0.81, 0.80, 0.83, 0.90, 0.85, 0.91, 0.93, respectively, and that the results

met the minimum allowable discrimination accuracy of 0.70 [36]. The overall accuracy of the decoding is more than 90%, which meets the research needs.

**Table 1.** Land-use data sources in the study area.

| Time | Image Types | Waveband | Wavelength | Resolution | Local Features | Land-use Features |
|------|-------------|----------|------------|------------|----------------|-------------------|
| 1966 | Panchromatic image of American keyhole satellite | panchromatic | | 2.7 m | | Mainly paddy field use |
| 1973 | 1:50,000 topographic map | | | 2.7 m | | Mainly paddy field use |
| 1990 | SPOT2 image | panchromatic | 0.50–0.73 μm | 10 m | | Mainly paddy field use |
| 2006 | SPOT5 image | panchromatic super-mode | 0.48–0.71 μm | 2.5 m | | Mainly paddy field use |
| 2010 | ALOS image | Band 1–3 | 1: 0.42–0.50 μm 2: 0.52–0.60 μm 3: 0.61–0.69 μm | 2.5 m | | Mainly modern agricultural land |
| 2016 | Google Earth image | Band 1–3 | 1: 0.45–0.52 μm 2: 0.52–0.60 μm 3: 0.63–0.69 μm | 1.7 m | | Mainly modern agricultural land |
| 2020 | GF-2 image | Band 1–3 | 1: 0.45–0.52 μm 2: 0.52–0.59 μm 3: 0.63–0.69 μm | 3.24 m | | Mainly modern agricultural land |

We classified the land-use status of the Lianjiang Basin into 10 categories of first-class land-use according to the current land-use classification standard [37] and the actual situation of the study area: cultivated land (1), forest land (3), grassland (4), industrial and mining storage land (6), residential land (7), public management and public service land (8), transportation land (10), water area and water area facilities land (11), other lands (12), and modern agricultural land (120). Modern agricultural land refers to facility agricultural land with a larger area and scale, such as greenhouse vegetable bases, flower bases, and economic fruit planting bases, with higher land-use intensification and efficiency. Combining the actual conditions of the study area and according to the current land-use classification promulgated in 2017 [37], we divided the second-class land-use status in the Lianjiang Basin area into 30 categories: paddy field (11), dry land (13), forested land (31), shrub land (32), natural grassland (41), industrial land (61), urban residential land (71), rural homestead (72), cultural and recreational land (85), highway land (102), street land (103), rural road (104), river water surface (111), pit water surface (114), ditches (117), free space (121), flower base (12011), landscape garden seedling base (12012), lawn planting base (12013), lotus root planting base (12021), greenhouse vegetable base (12022),

company contracted open-air vegetable base (12023), greenhouse strawberry planting base (12031), grape planting base (12032), dragon fruit planting base (12033), blueberry planting base (12034), greenhouse watermelon planting base (12035), economic fruit planting base (peach and plum) (12036) landscape ecological holiday leisure area (12041), and fish pond facility (12051) (Figure 3). The resolution of the remote sensing images in the study area at seven time periods was not consistent, so the authors conducted several field surveys for modification and validation, and the accuracy of the data reached more than 90%; on the other hand, the obvious changes in land-use in the study area occurred after 2000, so the difference in the resolution of the remote sensing images at seven time periods will not affect the results of this paper.

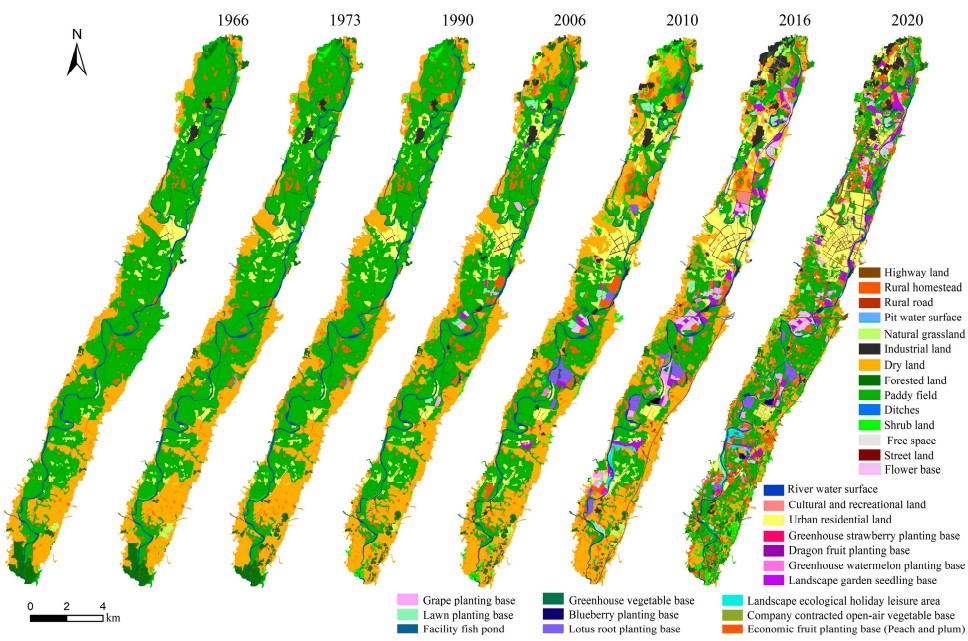

**Figure 3.** Land-use status for 7 periods in the Lianjiang Basin.

### 3.3. Research Scales and Calculation of Indicators

#### 3.3.1. Landscape Pattern Index

The Landscape Pattern Index is a tool for highly condensed and quantitative expression of landscape information. Therefore, based on the land-use data of the study area, this paper selects two indicators at the patch type scale and landscape level scale in the landscape index for analysis. Three indicators, total area of landscape (TA), percentage of landscape (PLAND), and patch density (PD), were included on the patch type scale, and one indicator, Shannon's Diversity Index (SHDI), was selected on the landscape level scale. The specific descriptions of the selected landscape pattern indices are shown in Table 2 [38,39].

**Table 2.** Landscape pattern indexes and their ecological significance.

| Landscape Index | Ecological Meaning |
| --- | --- |
| TA/hm$^2$ | Indicates the total area of a certain type of patch |
| PLAND/% | Indicates the percentage of landscape area that a given type of patch area represents per unit area of the study area |
| PD/n/hm$^2$ | Indicates the number of patches of a certain patch type in the unit area of the study area, which can be used to reflect the fragmentation of the landscape; the larger the value, the more fragmented the landscape |
| SHDI | Indicates the diversity of landscape types; the richer the landscape elements, the higher the value of the index |

3.3.2. Overall Scale

Starting from the overall scale of the study area, we selected the TA, PLAND, and cultivated patch size composition as three indicators to explore the overall land-use changes in the study area.

3.3.3. Buffer Scale

We divided the study area into 33 buffer zones starting at the northernmost end of the basin and made 1 km spaced buffers to the south (Figure 1). We drew on the buffer zone gradient strip method [40]. The 33 buffer zones were divided into different gradients, such as suburban Guiyang, the county town, suburbs of the county town, and the countryside, from north to south according to the study basin location conditions. We calculated the change in PLAND of construction land, modern agricultural land, and the change in SHDI and PD for each buffer strip to explore the gradient change law of land-use transformation in the basin.

3.3.4. Moving Window Scale

We converted the seven periods of land-use data in the study area to 10 m raster data and used Frastats 4.2 software to calculate the PLAND of land-use types and the SHDI by selecting a moving window of 100 m size [41].

**4. Results Analysis**

*4.1. Quantity Transition Characteristics of Land-use*

4.1.1. Quantity Transition Characteristics of Land-use for the Overall Basin

The transition changes in the overall land-use quantity of the basin in the study area are mainly reflected in three aspects: decreased cultivated land, increased residential land, and modern agricultural land (Figure 4a). Since the 1960s, the Lianjiang Basin as a whole has still been dominated by cultivated land as the main land type. However, over the past 50 years, the amount of cultivated land has shown a decreasing trend, while all other types of land have increased, except for fluctuations in the area of forest land types. From 1966 to 1990, the cultivated land area of the basins did not change significantly (71.52 to 69.53 km$^2$); from 1990, the cultivated land area began to decrease. By the end of 2020, the cultivated land area of the basin was only 37.54 km$^2$, of which the largest decrease was from 2010 to 2016; at the same time, the growth of residential land and modern agricultural land during the study period was also the largest, and modern agricultural land has increased significantly since 2010.

We further analyzed the changes in the secondary land types of the basin; its land-use transition changes continue to be reflected in three aspects: (1) paddy fields decreased significantly, and dry land remained basically unchanged; (2) the landscape of modern agriculture, such as flowers and seedlings, increased significantly; (3) construction land, such as urban and rural settlements and industrial land, increased (Figure 4b). Land-use transformation is changing from one land-use pattern (including dominant form and recessive form) to another in the region over time, driven by socio-economic changes and innovations [42]. Our research found that the basin's land-use form was stable from 1966 to 1990; since 2006, the land-use status of the basin has changed significantly; that is, land-use transition has occurred.

4.1.2. Quantity Transition Characteristics of Land-use in Different Basin Buffer Zones

Studying the distribution of the basin from north to south, we find Gaozhen town, Heping town (the location of the county seat), Sandu town, and Haohuahong town. The distribution pattern of basin construction land in the middle is high, and in the north and south are low, but the amount of construction land in the north is obviously higher than that in the south. After 2006, due to the expansion of cities and towns and the construction of industrial parks in the central and northern regions, the percentage of construction land was significantly higher than that in the south; the proportion of construction land in the

23 and 27 buffer zones where the southern towns and towns are located was higher than that of others.

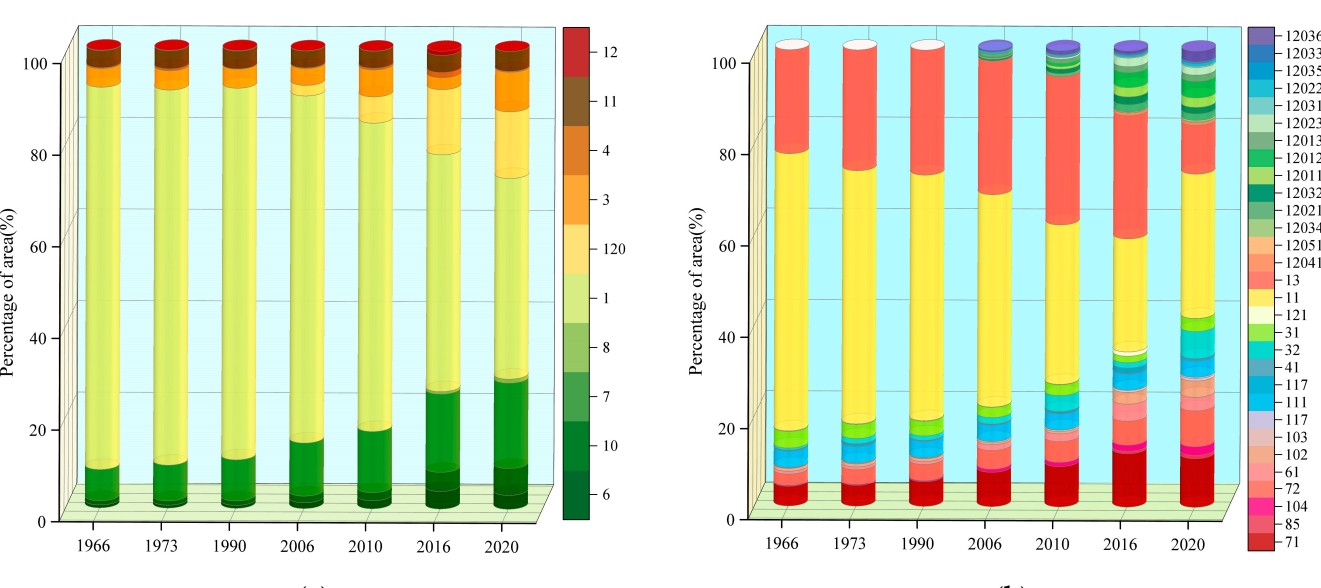

(**a**)                                                    (**b**)

**Figure 4.** (**a**) Area change of first-class land-use in the study area; (**b**) Area change of second-class land-use in the study area.

The distribution percentage of cultivated land across buffer zones is the opposite of the pattern of construction land distribution, with zones with a percentage of constructed land having a lower percentage of cultivated land. From 1966 to 2020, there was a general downward trend in cultivated land in all zones except for the southernmost 32 and 33 zones of the study basin, where there has been an increase in the area of cultivated land due to the conversion of forested land to cultivated land in recent years. In 2016, the percentage of cultivated land area was higher in the southern buffer zone of the basin than in the center and north.

In 2006, the cultivated land utilization function of the basin began to transform, and the percentage of modern agricultural land area, with various facilities, showed an increasing trend, which was mainly distributed around Gaozhen town in the north (4–11 strips), and from the county town in the middle to Sandu town (14–18 strips), as well as both sides of Haohuahong town in the south (20–30 strips). The change in the percentage of modern agricultural land area in the buffer zone indicates that the transition of cultivated land-use function in the study area expanded from the central and northern parts of the basin to the southern part of the basin (Figure 5).

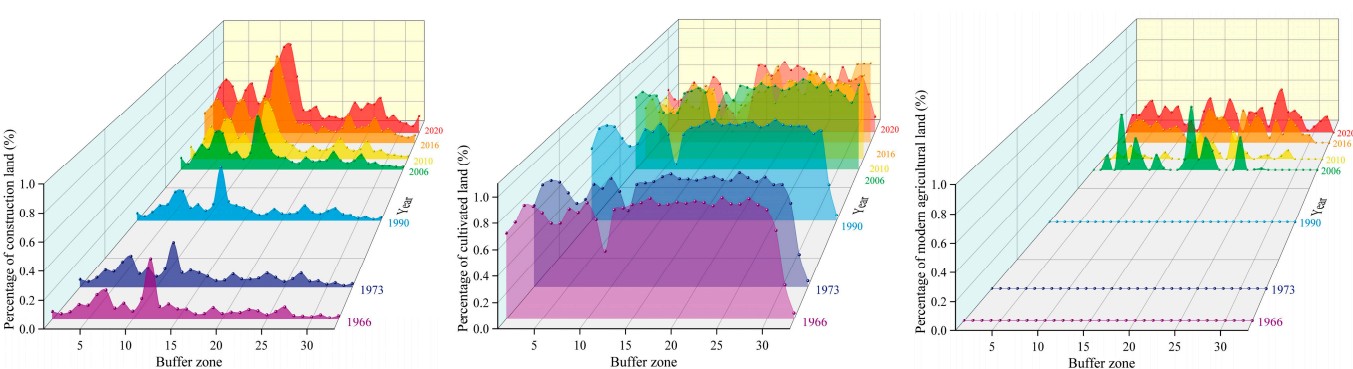

**Figure 5.** Land-use status in the different buffer ranges.

*4.2. Transition Characteristics of Land-use Type*

4.2.1. Type Transition Characteristics of Land-use for the Overall Basin

From the perspective of the transition of cultivated land in the basin, the change in cultivated land in the study area is mainly based on transfer out, with a small number of transfers in during some of the study periods. On the whole, before 1990, the cultivated land in the basin mainly transferred out in the direction of forest land and residential land. After 1990, the area and the magnitude of change of cultivated land converted to industrial, mining, storage, transportation, and residential land, etc., gradually increased, and began to be converted to modern agricultural land, with the magnitude of the transfer increasing rapidly over time, and part of the forest land was converted to cultivated land during this period (Figure 6).

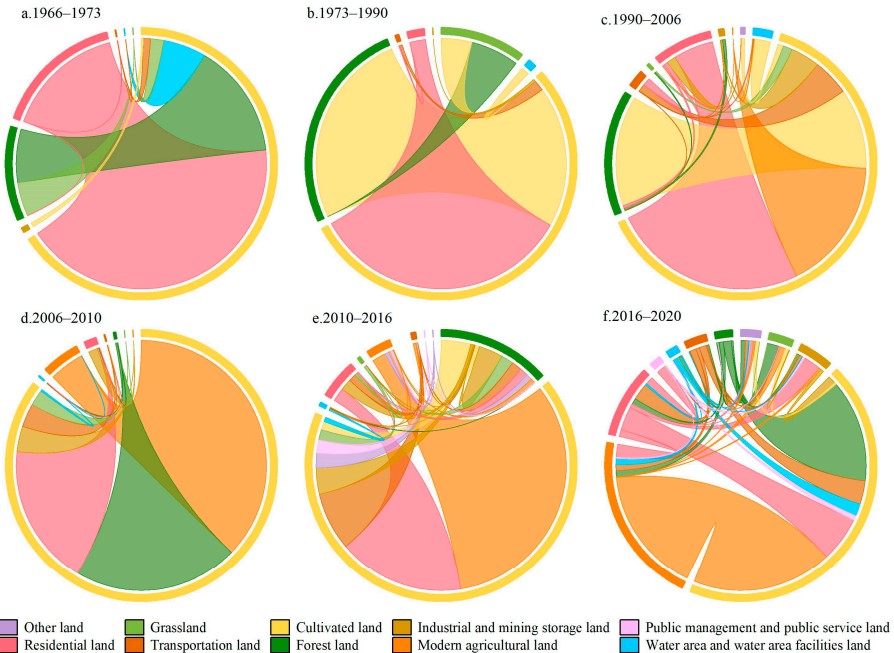

**Figure 6.** Transfer matrix of the first-class land-use type in the study area.

Specifically, from 1966 to 1973, the main direction of the transfer of cultivated land in the basin was towards residential land (urban residential land and rural homestead), with a transfer ratio of 64.56%, followed by forest land (forested land and shrubland), with a ratio of 22.67%. From 1973 to 1990, the total area transferred out decreased, and the proportion of forest land converted into cultivated land increased by 77.50%. From 1990 to 2006, the diversified direction of cultivated land transferring out and the amplitude of transferring out also increased gradually and, especially, the transfer to industrial, mining and storage land, transportation land, modern agricultural land, and residential land accounted for 5.98%, 7.50%, 27.06%, and 39.95%, respectively; among these, the regional space of the change in the direction of the transfer to residential land was mainly distributed in the area around Gaozhen town in the north of the basin and around the periphery of Huishui county in the central part of the basin, and the change in the direction of the transfer to modern agricultural land-use, on the other hand, is mainly concentrated in the vicinity of Xiaolongba, north of Sandu town.

The main transfer direction of cultivated land from 2006 to 2010 continued to be focused on modern agricultural land, accounting for 43.58%. Still, the proportion of land transferred to residential land decreased to 21.40% in that period, and the proportion transferred to forest land increased to 24.94%. Still, local greenhouse watermelon plantation land was restored to traditional cultivated land in that period. From 2010 to 2016, the transfer of cultivated land in the basin showed the most diversification, it was transferred to

all land types, and the conversion rate increased rapidly (of which the modern agricultural land direction is the largest proportion of conversion, at 49.46%, followed by 24.36% of residential land and 11.255% of transportation land); the sources of transfer into forest, modern agricultural land and residential land showed proportions of 53.22%, 19.35%, and 19.01%, respectively. Under the implementation of the policy of cultivated land protection and food security from 2016 to 2020, the area of cultivated land in the basin showed a trend of recovery, and the recovery of cultivated land mainly originated from modern agricultural land, accounting for 70.78%, followed by residential land, accounting for 13.14%; the direction of transferring out in the area is still dominated by modern agricultural land, accounting for 41.46%, followed by forest land and residential land, accounting for 30.31% and 13.39%, respectively.

4.2.2. Type Transition Characteristics of Land-use in Different Basin Buffer Zones

This can be obtained from Figure 7. The basin area was mainly converted from paddy fields to construction land, and the space was scattered from 1966 to 1973. Between 1973 and 1990, paddy fields were transferred to dry land, mainly around the towns. The transfer of paddy land to construction land in 1990–2006 was concentrated around the towns of Heping and Gaozhen in the northern part of the basin in the form of more concentrated and contiguous large patches. From 2006 to 2010, the loss of paddy fields and dry land began to increase significantly; from 2010 to 2016, the loss of dry land in the basin's northern, central, and southern parts was concentrated. The loss of dry land in 2016–2020 occurred mainly at the edge of the basin, in the north and the south, and the loss of paddy land mainly in the south-central part of the basin.

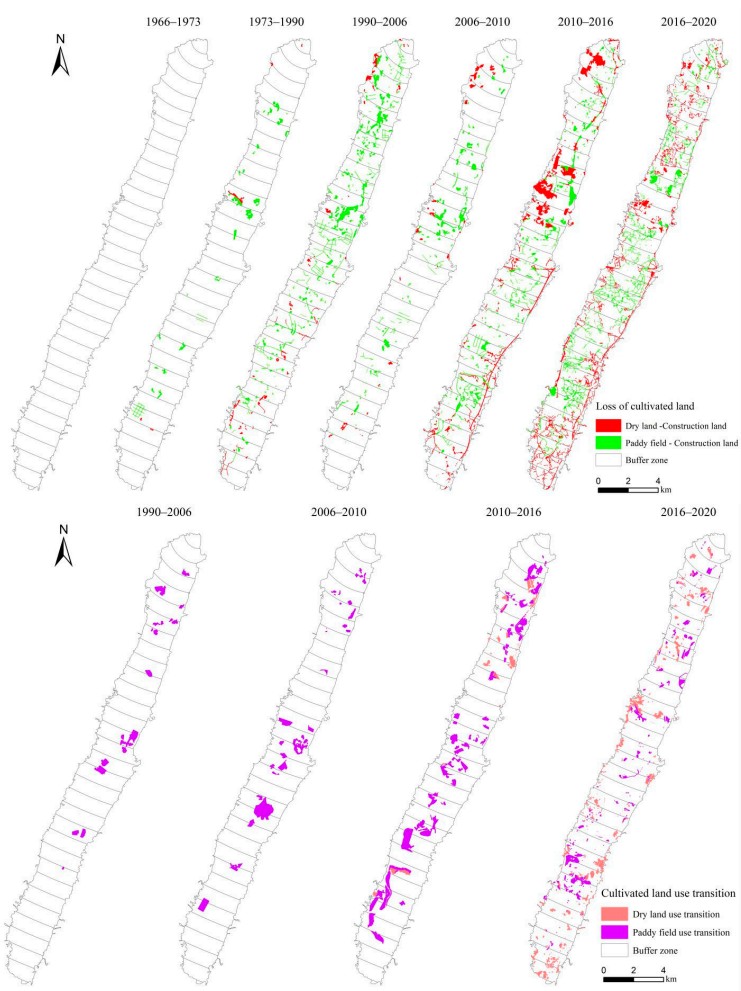

**Figure 7.** Spatial pattern of land-use transfer direction in the study area.

In the four time periods from 1990 to 2020, the transition of cultivated land-use in the basin was dominated by the transformation of paddy field use. Cultivated land transformation in 1990–2006 was distributed in Gaozhen town in the north and Sandu town in the middle; in 2006–2010, the transformation of cultivated land in the center increased significantly, and it was mainly converted to vegetable land in these two time periods. From 2010 to 2020, the transformation of cultivated land-use basically expanded to the entire basin, converted primarily into modern agricultural land for flower and landscape garden seedling bases.

### 4.3. Landscape Pattern Transition Characteristics of Land-use

4.3.1. Landscape Transition Characteristics of Land-use for the Overall Basin

Before 2006, the area of paddy fields and dry land in the basin was greater than 100 ha and 50–100 ha, respectively; after 2006, the area was mainly 20–50 ha (Figure 8). Paddy fields dominate cultivated land in the study area, and the spatial distribution of the percentage area composition of paddy patches and the percentage in the unit area showed that the degree of fragmentation of paddy fields increased significantly in 2006 and later. With the evolution of land-use in the basin, cultivated land is gradually occupied and transformed, which not only led to the fragmentation of cultivated land patches and the evolution from a centralized to a fragmented pattern (Figure 9), but also led to the land-use diversity index moving from low to high, and the land-use pattern becoming gradually more complex (Figure 10).

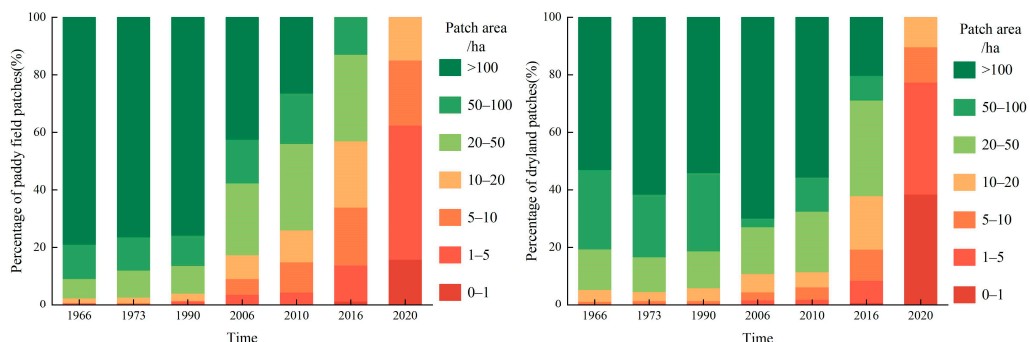

**Figure 8.** Constituent of cultivated land patches in the study area for 7 periods.

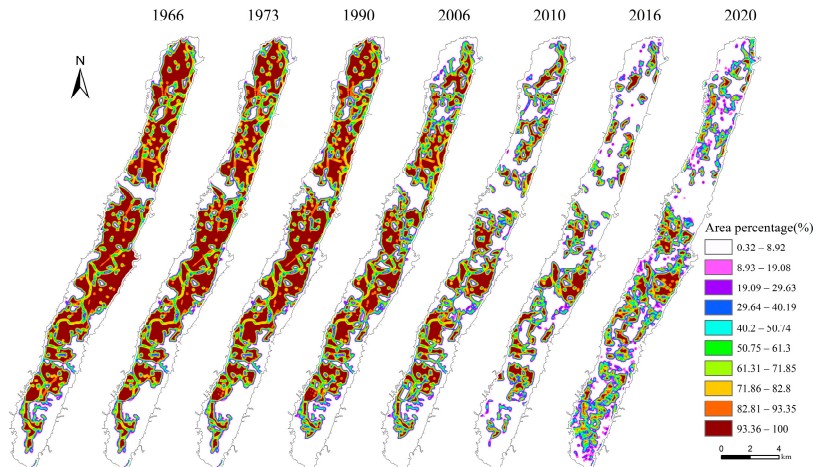

**Figure 9.** PLAND of the paddy field in the study area for 7 periods.

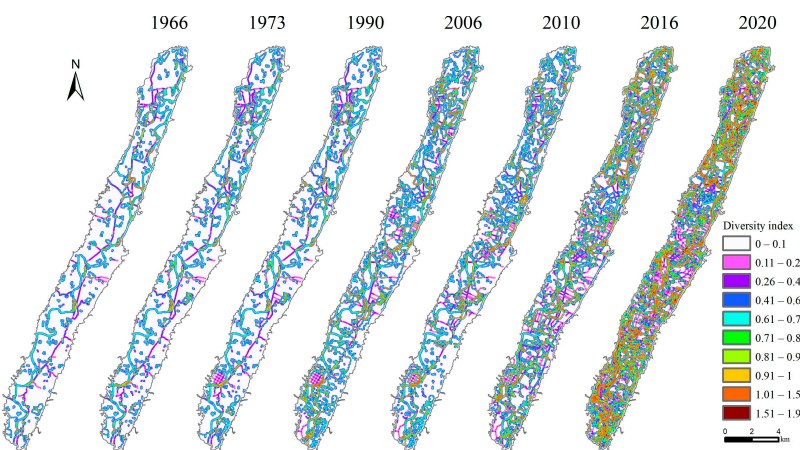

**Figure 10.** Spatial distribution of land-use diversity index in the study area for 7 periods.

4.3.2. Landscape Transition Characteristics of Land-use in Different Basin Buffer Zones

This paper selects the number of modern agricultural land types, PD, and SHDI of each buffer zone to reflect each buffer zone's landscape characteristics and changes (Figure 11). The change curves of the number of modern agricultural land types and the PD of modern agricultural land are basically consistent with the change patterns of the percentage of modern agricultural land in each buffer zone, and the change patterns of the percentage of construction land and cultivated land in each buffer zone show roughly opposite trends.

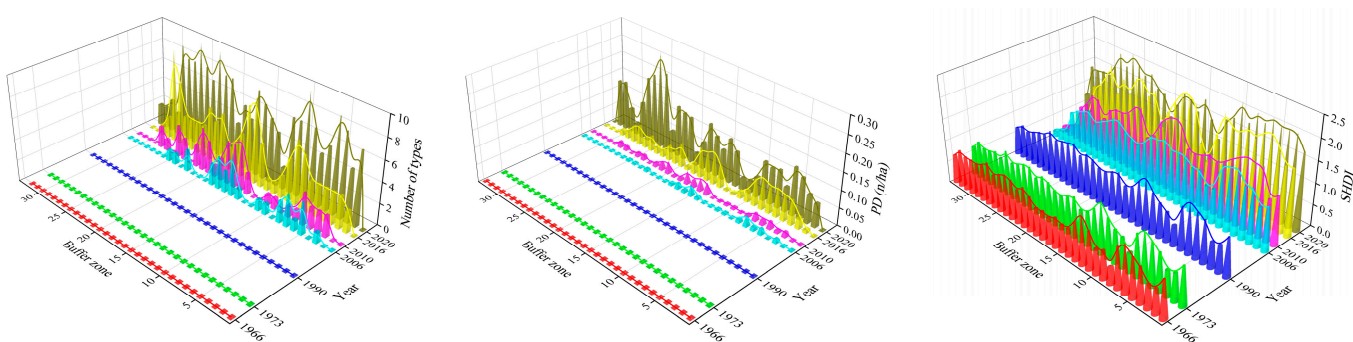

**Figure 11.** The number of modern agricultural uses, the PD, and the SHDI.

The change in the land-use diversity index in each buffer zone is a comprehensive reflection of the combination pattern of construction land, cultivated land, and modern agricultural land in each buffer zone. The 12th buffer zone is the location of the Huishui county seat, and the land-use diversity index is significantly lower than that of the nearby buffer zone; the 32nd and 33rd buffer zones at the southernmost end of the basin are still used for traditional agriculture in 2016, and the land-use is single. Therefore, their land-use diversity index was lower than that of other buffers in 2016. Strips with a great variety and proportion of modern agricultural land and buffer strips with a high proportion of construction land, such as north of Gaozhen town, generally have a high land-use diversity index due to the fragmentation of the cultivated landscape by modern agricultural land and construction land. From the time perspective, the changes in each buffer zone's land-use diversity index are consistent with the spatial distribution of the land-use diversity index in the basin. Their land-use diversity indices mainly show an increase, except for the 12th buffers and 32nd and 33rd buffer zones.

*4.4. Gradient Effects of Land-use Transition in the Basin*

Because the north study basin is close to Guiyang city, according to the location conditions, we can divide it into different gradients, such as suburban Guiyang, the county town, suburbs of the county, and countryside from south to north, and land-use transition

from north to south showed the change in the gradient: "industrial land-urban construction land-modern agricultural land-traditional agricultural land" (Figure 12).

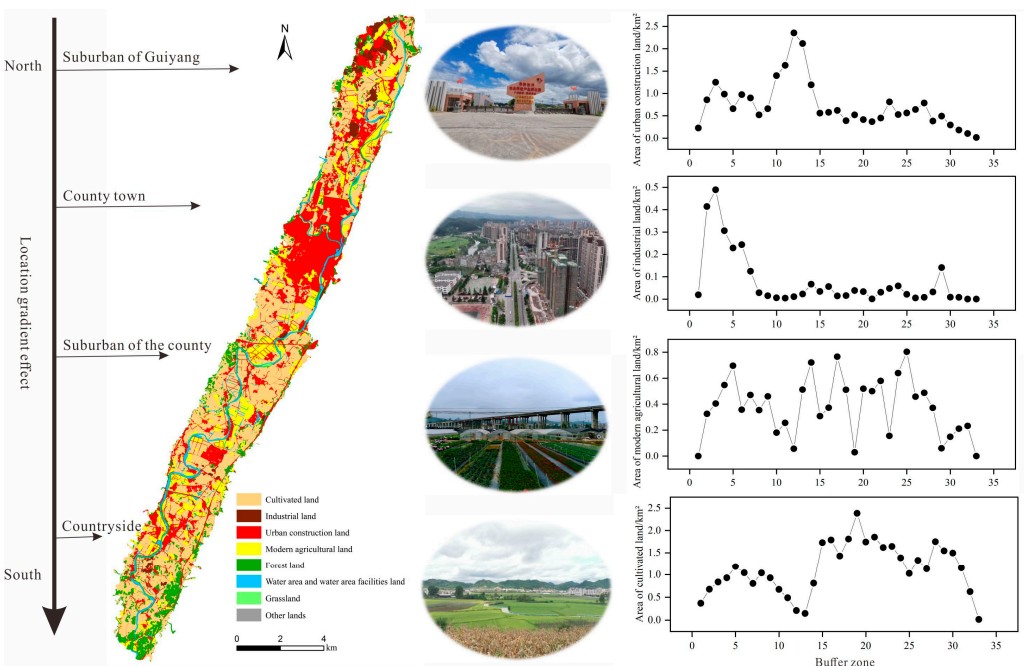

**Figure 12.** Gradient effects of land-use transition in the basin.

Specifically in the buffer zone, industrial land is mainly distributed in 2–4 strips around suburban Guiyang, showing an inverted "V" trend of "upward-decline." The urban construction land is primarily concentrated in Huishui County, showing a normal distribution from north to south, but the distribution area in the northern Guiyang suburb is significantly higher than in the suburban county and countryside areas in the south. The cultivated land is mainly concentrated in the southern part of the study area, and the trend in cultivated land area and urban construction land area showed an opposite change on each strip. Modern agricultural land is distributed in a "jagged" pattern from north to south, with high values mainly located in the suburban Guiyang, the suburban county, and in the vicinity of the village of Haohuahong in the south.

## 5. Discussion

### 5.1. Theoretical Model of Evolution and Transformation of Land-use in Rural Basins in Karst Mountainous Areas

With the development of human society, land-use is transforming to the stage of gradual intensification and intensive use [1,43]. Currently, most developing countries are in the stage of social transition and land-use is moving towards intensive development [10,44]. In China, the process of rural transition and development is mainly reflected in cultivated land-use transition and rural homestead use transition [45].

Against the background of rapid urbanization and vigorous promotion of agricultural transition and upgrading in Guizhou Province, in addition to the main function of providing agricultural and sideline products, improving ecological/landscape/social functions and realizing multi-functional agriculture have become important tasks for the rural basin's development. Against this background, the cultivated land, modern agricultural land, forest land, and other agricultural lands in the basin will inevitably undergo reorganization and transition under the influence of market, planning, and natural conditions; at the same time, the non-agricultural production and non-productive functions of the basin will also change accordingly [29]. According to the evolution trends and landscape pattern changes for the first-class and second-class types of land in the basin, we divided the research stage

of land-use change for the basin into four stages: the traditional grain production stage, the mixed development stage of traditional grain production and greenhouse cultivation, the modern and efficient agricultural production stage and the market-oriented comprehensive agricultural development stage, and the overall trend in land-use change is the transition from traditional rural landscapes to mixed landscapes (Figure 13). This paper validates the proposed hypothesis, which reflects the general characteristics of land-use change in the mountainous "granary" and conforms to the future transition direction of Chinese agriculture, where large-scale bulk agriculture and specialized fine agriculture coexist [46]. Basins of more than 1000 mu in Guizhou Province include paddy field basins, dryland basins, urban industrial basins, other land-use basins, etc. Larger basins in Guizhou tend to be the locations of cities, counties, and towns and generally face similar land-use changes and cultivated land-use transitions as those in the basins in the study area. Some studies have shown that comparative interests drive the transition of slope land-use in mountainous areas, and the slope-cultivated land has evolved into diversified landscapes such as economic and fruit forest land, construction land, etc., realizing ecological and economic win–win benefits [47]. In other plains, some local governments have encouraged farmers to grow large quantities of non-grain products to achieve rapid economic development [48,49], leading to a shrinking area under food cultivation, which may eventually lead to food security problems. Therefore, the government should strictly prohibit non-food production on basic farmland, foster new agricultural business entities, develop local specialty food crops, and formulate relevant food security protection policies based on local conditions.

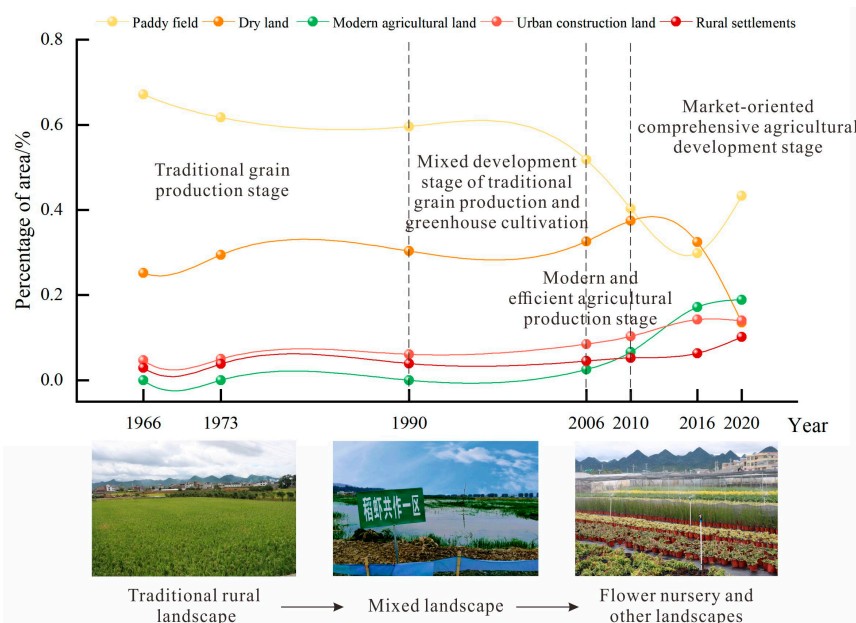

**Figure 13.** Theoretical model of land-use change and transition in the study basin.

### 5.2. Driving Mechanism of Land-use Transition in the Basin

The land-use of the basin in the karst mountains is affected more by the conditions of the basin itself and other social and economic factors, such as towns, traffic conditions, and location factors. In turn, with the development of urbanization in the karst mountains, improving road conditions and changing location conditions will inevitably cause changes in the basin's land-use function. In the basin, traffic, location conditions, and land resource conditions have the most significant impact on the spatial location where the land-use transition occurs, which is the direct driving factor leading to the study of land-use transition in the basin. The cultivated land loss and cultivated land transformation first occurred in the northern and central regions close to the provincial capital and county, and at the same time they were mainly distributed on both sides of the provincial road from the central to the north and south of the basin. Some terraces and hills in the inner and at the edge of

the basin make the terrain undulate. The transformation of cultivated land-use avoids the hilly terraces and is distributed in the Lianjiang Basin with flat terrain, concentrated and contiguous fertile soil, and convenient irrigation.

Land-use type conversion has resulted from multiple factors of the natural environment and social economy [50,51]. In the process of urbanization and industrialization, the transformation of population and industry plays a decisive role in the intensity and direction of the evolution of cultivated land's economic, social, and ecological functions [52]. For the basin, urban radiation, market demand, profitabilization of land output, rural tourism development, foreign capital injection, and government promotion have jointly promoted the adjustment of the agricultural structure in the study area. All this promotes the large-scale, specialized, economic, and ecological utilization of cultivated land [12,53], creates a multi-functional landscape, and forms experiential, sightseeing, and comprehensive modern leisure agriculture, jointly promoting the modernization and transformation of rural areas (Figure 14).

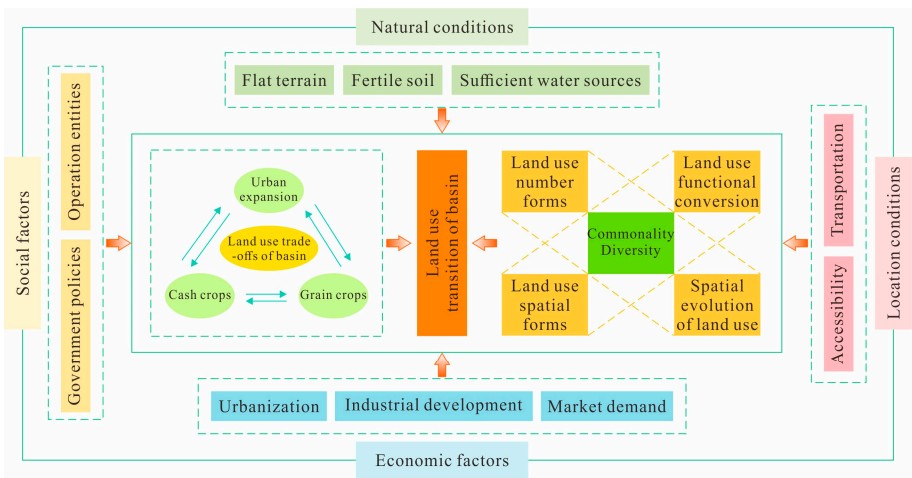

**Figure 14.** Driving mechanisms of land-use transformation in the basin.

### 5.3. Effect and Enlightenment of Land-use Transition in the Basin

In recent years, agricultural production has undergone a transition, which has brought forward a more urgent need for research on the transition of cultivated land-use [54,55]. How are we to guide the land-use transition to adapt to the local natural environmental conditions and the regional economic and social development stage, and to make this happen in an appropriate area [56,57], putting forward higher requirements for land management? As far as the mountainous "granary" and Guizhou are concerned, the basin's land management policies and systems need to consider the land-use transition, the current land-use transition stage, and the corresponding stage of regional socio-economic transition [58]. Further, the study of the hidden land-use form of the basin and its changes should be the focus of future rural revitalization through the control of land-use transition [59,60]. Moreover, how can the remaining traditional agricultural space be managed regarding the transition of land-use in the basin? How can we properly control the land-use transition regarding the high natural value and high traditional agricultural value of basin? The ecological and environmental effects of basin land-use transition should also receive attention.

Meanwhile, in Guizhou Province, dominated by mountainous areas, cultivated land was mainly converted to forest land after 1990 [61], but the land-use of the mountainous regions and basin also had a certain coupling effect [26]. Therefore, the study area also influences the change in land-use around the basin, forming the coupled evolution of mountain-basin land-use, which needs further study. Many factors affect China's food security, including the rapid transition of arable land [62]. The change in cultivated land pattern has a significant impact on cultivated land system security and is the foundation for ensuring adequate provision of cultivated land and the premise of achieving sustain-

able development for society [63]. In turn, guaranteeing regional food security requires ensuring the sustainability of cultivated land utilization [64] and stabilizing cultivated land areas [65]. Within the context of rapid socio-economic development and the diversification of the population's demand for agricultural products, the utilization of cultivated land in "granary" basins in karst mountainous areas is a trade-off between regional socio-economic development, ecological protection, and food security. Therefore, studying the spatial and temporal evolution of land-use characteristics and transition patterns in mountainous "granary"-type basins at different scales for optimization of regional land-use, regional food security, and rural revitalization is significant. Further work will collect detailed socio-economic data and conduct an in-depth study on the coupling and harmonization of human–land relations within the "granary" basins, combining the conditions of social economy, population, natural environment, and land-use.

## 6. Conclusions

Based on the theoretical analysis of the land-use transition in the mountainous "granary" in southwest China, this paper takes a typical basin, the Lianjiang Basin, as an example of a typical granary basin and utilizes the high-resolution data of seven periods from 1966 to 2020, combined with field investigation, and discusses land-use spatial evolution and the land-use transition rule of the basin in the last 50 years at a multi-spatial scale, from the perspective of quantity change, the perspective of type change, and the perspective of landscape pattern change. The results show that land-use transition in the basin mainly includes the transfer of cultivated land functions and the diversification of cultivated land functions. Cultivated land is gradually occupied and transformed, resulting in the fragmentation of cultivated land, from continuous patch to fragmentation pattern, and the diversity of land-use increases. The stage of land-use change in the study basin is divided into four phases: the traditional grain production stage, the mixed development stage of traditional grain production and greenhouse cultivation, the modern and efficient agricultural production stage, and the market-oriented comprehensive agricultural development stage, and the landscape pattern is also gradually transformed from the traditional rural landscape to the modern agricultural landscape. The gradient effect of land-use evolution and transformation in the study basin was obvious, with the functional transformation of basin cropland-use expanding from the central and northern parts of the basin to the southern part of the basin. From 2010 to 2016, the transition of cultivated land utilization basically extended to the whole research area, which mainly turned into modern agricultural land. Traffic, location, and land resources significantly affect land-use transformation in the basin. From this, we put forward several suggestions on the optimization of land-use in such basins; firstly is the implementation of the responsibility for the protection of cultivated land; cultivated land in such basins is the essence of the land resources in mountainous areas, and it is necessary to clarify the protection tasks of cultivated land in basins, especially paddy fields. Secondly, the requisition–compensation balance of cultivated land should be reformed and improved, and the occupation of high-quality paddy land within basins should be strictly prohibited. Thirdly, a compensation mechanism for food growing has been developed to mobilize farmers to grow food. Fourthly, local managers should explore sustainable development models for the basin area to improve land-use intensification, such as crop rotation between grain and non-grain, and rice-crayfish culture. Further work will collect detailed socio-economic data and conduct an in-depth study on the coupling and harmonization of human-land relations within the "granary" basins, combining the conditions of social economy, population, natural environment, and land-use.

**Author Contributions:** Conceptualization, M.C. and Y.L.; methodology, M.C., Y.L., Y.Z. and L.Y. (Limin Yu); software, M.C., Y.L., L.Y. (Limin Yu) and L.Y. (Linyu Yang); validation, M.C., Y.L., Y.Z., L.Y. (Limin Yu) and L.Y. (Linyu Yang); formal analysis, M.C., Y.L., Y.Z. and L.Y. (Linyu Yang); investigation, M.C., Y.L., L.Y. (Limin Yu) and L.Y. (Linyu Yang); resources, M.C., Y.L., Y.Z., L.Y. (Limin Yu) and L.Y. (Linyu Yang); data curation, Y.Z., Y.L. and L.Y. (Limin Yu); writing—original draft preparation, M.C. and Y.L.; writing—review and editing, M.C., Y.L. and Y.Z.; visualization, M.C., Y.L., Y.Z. and L.Y.

(Linyu Yang); supervision, M.C., Y.L., L.Y. (Limin Yu) and L.Y. (Linyu Yang); project administration, M.C. and Y.L.; funding acquisition, Y.L. All authors have read and agreed to the published version of the manuscript.

**Funding:** This research was funded by the National Natural Science Foundation of China, grant numbers 41661020 and 42061035, and by Guizhou Science and Technology Cooperation Platform Talents [2021] A22.

**Data Availability Statement:** Data are contained within the article.

**Conflicts of Interest:** The authors declare no conflict of interest. The funders had no role in the design of the study; in the collection, analyses, or interpretation of data; in the writing of the manuscript; or in the decision to publish the results.

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
