# Peer review of "Evolution and Transformation Analysis of Land-use in Mountainous “Granary”—Evidence from Typical Basin in Karst Mountainous Areas of Southwest China"

_land, doi:10.3390/land13010004_

Round 1

Reviewer 1 Report

Comments and Suggestions for Authors

Revealing the process and trend of land use change in intermountain basins is of great significance for optimizing land use, food security, and rural development in mountainous areas. This article provides a very detailed and in-depth analysis of land use in the Lianjiang Basin of Huishui County, Guizhou Province, China. On the basis of high-resolution remote sensing images and field investigations, the analysis methods of landscape pattern, moving window, and buffer zone were used to deeply study the spatial evolution and transformation laws of land use from the perspectives of changes in land use quantity and type, landscape pattern changes, etc. It presents the trends and details of land use change in the region in a convincing way. Meanwhile, the article also analyzed the significant impacts of transportation, location, and land resources on the transformation of watershed land use.

The manuscript is well written. The work is well related to previous work in the field. One little point is that the figures are too indistinct.

Author Response

Dear Reviewers,

Thank you for taking the time out of your busy schedule to read and modify my manuscript (Manuscript ID: land-2758532). You have given us many constructive comments on our manuscripts, which have greatly improved the quality of the paper. We have carefully revised the manuscript according to the reviewer's suggestion and hope to meet with approval. Please see the attachment for the major revisions to the paper, responses to reviewer comments, and some comparative examples of before and after revisions.

Reviewer 2 Report

Comments and Suggestions for Authors

Review: Evolution and Transformation Analysis of Land Use in Mountainous "Granary"-Evidence from Typical Basin in Karst Mountainous Areas of Southwest China

Dear authors and editors,

First, I would like to express my congratulations for touching on an issue, namely in-depth study on the spatial evolution of land use and land use transformation. By adopting selected analysis methods, the perspectives of changes in the quantity of land use, changes in the types of land use, and changes in the landscape pattern have been explored in the study. Second, the manuscript is also interesting for a reader because of the successful discussion of proper to study practical achievements. The authors demonstrate a range of literature that underlines the importance of the topic based on previous studies. Thus, I believe that this could be a pertinent, valuable and even necessary paper. However, it is also evident from the study that it is very comprehensive and some sub-sections, e.g. 2.3, do not give enough clear understanding of the written, and thus it is difficult to follow the research logic.

The quality of the manuscript should be improved. Thus, it is advised to consider the following:

1.     The manuscript consists of many long sentences, which makes it difficult to read and understand the text. For instance, the objectives of the paper should be clearly set. Therefore, the last paragraph of Introduction (Section 1) should be subdivided into several sentences (see lines 103-111).

2.     Many terms and concepts are not clearly explained and indicators used in the study, perhaps proper references are necessary to indicate. For instance, modern agricultural space, sightseeing agriculture, patch density, gradient change law, etc. (see lines 200-202 and others).

3.     What are the commonalities or differences between – second level land use, secondary land types, and second-class land use? All these appear in different places of the study, but their status/meaning is not clear to the reviewer.

4.     Perhaps, trade-offs between socio-economic development, ecological protection and food security could be better explained by effects and their dynamics showing the values of particular indicators. Thus, the Section 4 (Discussion) appears quite descriptive and general.

5.     Conclusions (Section 5) should include more specific suggestions (see the last paragraph), as well as this section should respond to the set objectives of the study and indicate recommendations to the decision-making authorities regarding land use optimisations in the basins.

Given all the above-mentioned, I recommend to the editors that MAJOR REVISION be addressed before accepting the paper for publishing.

I wish you the best of luck with your research and thank you again for conducting this interesting discussion.

With best regards

The reviewer

Author Response

(The authors gave the same response as above.)

Reviewer 3 Report

Comments and Suggestions for Authors

The paper's topic is generally interesting. Its most compelling aspect lies in the analysis of various remote sensing products using visual interpretation. This method, although time-consuming, significantly aids researchers in producing accurate LULC maps. However, the paper requires further improvement, as outlined in my comments within the relevant sections.

Please avoid using overly long sentences and work on improving the overall sentence structure throughout the paper to enhance clarity and coherence. for example line 15-21.

The abstract is well-written and structured. To enhance it further, consider incorporating more discussion within this section. Additionally, improving the methodology section, especially regarding the utilized datasets, would be beneficial.

In the first part of the introduction, you discuss the concepts of LULC transformation. Following this, statistical data from the USA and China is presented. However, there is a disjointed flow as additional sentences follow, introducing more statistics without cohesive connections between them. This section needs revision to ensure smoother transitions and better linkage between sentences.

In the concluding segment of the introduction, it's essential to clearly outline the research objectives and questions.

Incorporate spectral bands and utilized features into Table 1 to provide a comprehensive overview of the data employed in the study.

In the methods section, it's important to elucidate the rationale behind choosing visual interpretation over automatic image classification methods. Describe the validation process employed and detail how the extracted LULC maps were validated for accuracy and reliability. This explanation will provide insights into why this particular approach was chosen and establish the credibility and trustworthiness of the generated LULC maps.

The section 2.3.1. Overall scale, is too short. add more explanation.

Comments on the Quality of English Language

The paper could benefit from improvements in language to enhance its clarity and readability. see my comments.

Author Response

(The authors gave the same response as above.)

Round 2

Reviewer 2 Report

Comments and Suggestions for Authors

Dear Authors and Editors

In view of provided responses and revision of manuscript, I recommend to the Editors accepting the paper for publishing.

Wishing you the best of luck with your research, and thanking you again for conducting this interesting discussion.

With best regards

The reviewer